# Fairness without Demographics through Adversarially Reweighted Learning

**Preethi Lahoti** *
plahoti@mpi-inf.mpg.de
Max Planck Institute for Informatics

**Alex Beutel, Jilin Chen, Kang Lee, Flavien Prost,**
**Nithum Thain, Xuezhi Wang, Ed H. Chi**
Google Research

## Abstract

Much of the previous machine learning (ML) fairness literature assumes that protected features such as race and sex are present in the dataset, and relies upon them to mitigate fairness concerns. However, in practice factors like privacy and regulation often preclude the collection of protected features, or their use for training or inference, severely limiting the applicability of traditional fairness research. Therefore we ask: *How can we train an ML model to improve fairness when we do not even know the protected group memberships?* In this work we address this problem by proposing Adversarially Reweighted Learning (ARL). In particular, we hypothesize that non-protected features and task labels are valuable for identifying fairness issues, and can be used to co-train an adversarial reweighting approach for improving fairness. Our results show that ARL improves Rawlsian Max-Min fairness, with notable AUC improvements for worst-case protected groups in multiple datasets, outperforming state-of-the-art alternatives.

## 1 Introduction

As machine learning (ML) systems are increasingly used for decision making in high-stakes scenarios, it is vital that they do not exhibit discrimination. However, recent research [19, 5, 30] has raised several fairness concerns, with researchers finding significant accuracy disparities across demographic groups in face detection [9], health-care systems [22], and recommendation systems [15]. In response, there has been a flurry of research on fairness in ML, largely focused on proposing formal notions of fairness [19, 20, 54, 20], and offering "de-biasing" methods to achieve these goals. However, most of these works assume that the model has access to protected features (e.g., race and gender), at least at training [55, 13], if not at inference [20, 24].

In practice, however, many situations arise where it is not feasible to collect or use protected features for decision making due to privacy, legal, or regulatory restrictions. For instance, GDPR imposes heightened prerequisites to collect and use protected features. Yet, in spite of these restrictions on access to protected features, and their usage in ML models, it is often imperative for our systems to promote fairness. For instance, regulators like CFBP require that creditors comply by fairness, yet prohibit them from using demographic information for decision-making.[2] Recent surveys of ML practitioners from both public-sector [50] and industry [24] highlight this conundrum, and identify "addressing fairness without demographics" as a crucial open-problem with high significance to ML practitioners. Therefore, in this paper, we ask the research question:

*How can we train a ML model to improve fairness when we do not have access to protected features neither at training nor inference time, i.e., we do not know protected group memberships?*

**Goal:** We follow the Rawlsian principle of *Max-Min* welfare for distributive justice [46]. In Section 3.1, we formalize our *Max-Min* fairness goals: to train a model that maximizes the minimum expected utility across protected groups with the additional challenge that, *we do not know protected group memberships*. It is worth noting that, unlike parity based notions of fairness[20, 54], which aim to minimize gap across groups, *Max-Min* fairness notion permits inequalities. For many high-stakes ML applications, such as *healthcare* and *face recognition*, improving the utility of worst-off groups is an important goal, and in some cases, parity notions that equally accept decreasing the accuracy of better performing groups are often not reasonable.

**Exploiting Correlates:** While the system does not have direct access to protected groups, we hypothesize that unobserved protected groups $S$ are correlated with the observed features $X$ (e.g., race is correlated with zip-code) and class labels $Y$ (e.g., due to imbalanced class labels). As we will see in Table 4 (§5), this is frequently true. While correlates of protected features are a common cause for concern in the fairness literature, we show this property can be valuable for *improving* fairness metrics. Next, we illustrate how this correlated information can be valuable with a toy example.

*Illustrative Example:* Consider a classification task wherein our dataset consists of individuals with membership to one of the two protected groups: "orange" data points and "green" data points. The trainer only observes their position on the $x$ and $y$ axis. Although the model does not have access to the group (color), $y$ is correlated with the group membership.

Although each group alone is well-separable (Figure 1(a-b)), we see in Figure 1(c) that the empirical risk minimizing (ERM) classifer over the full data results in more errors for the green group. Even without color (groups), we can quickly identify a region of errors with low $y$ value (bottom of the plot) and a positive label (+). In Section 3.2, we will define the notion of computationally-identifiable errors that correspond to this region. These errors are in contrast

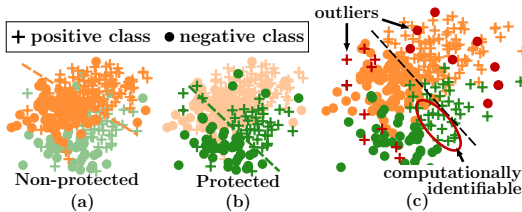

Figure 1: Computational-identifiability example

to outliers (e.g., from label noise) with larger errors randomly distributed across the $x$-$y$ axes.

The closest prior work to ours is *DRO* [21]. Similar to us, *DRO* has the goal of fairness without demographics, aiming to achieve Rawlsian Max-Min Fairness for unknown protected groups. However, to achieve this, *DRO* uses distributionally robust optimization to optimize for the worst-case groups by focusing on improving any worst-case distributions, but as the authors point out, this runs the risk of focusing optimization on noisy outliers. In contrast, we hypothesize that focusing on addressing computationally-identifiable errors will better improve fairness for the unobserved groups.

**Adversarially Reweighted Learning:** With this hypothesis, we propose Adversarially Reweighted Learning (ARL), an optimization approach that leverages the notion of computationally-identifiable errors through an adversary $f_\phi(X, Y)$ to improve worst-case performance over unobserved protected groups $S$. Our experimental results show that ARL achieves high AUC for worst-case protected groups, high overall AUC, and robustness against training data biases.

Taken together, we make the following contributions:

- **Fairness without Demographics:** In Section 3, we propose adversarially reweighted learning (*ARL*), a modeling approach that aims to improve the utility for worst-off protected groups, without access to protected features at training or inference time. Our key insight is that when improving model performance for worst-case groups, it is valuable to focus the objective on *computationally-identifiable* regions of errors.
- **Empirical Benefits:** In Section 4, we evaluate *ARL* on three real-world datasets. Our results show that *ARL* yields significant AUC improvements for worst-case protected groups, outperforming state-of-the-art alternatives on all the datasets, and even improves the overall AUC on two of three datasets.
- **Understanding ARL:** In Section 5 we do a thorough experimental analysis and present insights into the inner-workings of ARL by analyzing the learnt example weights. In addition, we perform a synthetic study to investigate robustness of *ARL* to worst-case training distributions. We observe that *ARL* is quite robust to representation bias, and differences in group base-rate. However, similar to prior approaches, *ARL* degrades with noisy ground-truth labels.

## 2   Related Work

**Fairness:**  There has been an increasing line of work to address fairness concerns in machine learning models. A number of fairness notions have been proposed. At a high level they can be grouped into three categories, including (i) individual fairness [13, 56, 38, 37], (ii) group fairness [16, 32, 20, 55] that expects parity of statistical performance across groups, and (iii) fairness notions that aim to improve per-group performance, such as Pareto-fairness [4] and Rawlsian Max-Min fairness [46, 21, 58, 43]. In this work we follow the third notion of improving per-group performance.

There is also a large body of work on incorporating these fairness notions into ML models, including learning better representations [56, 38, 37] and adding fairness constraints in the learning objective [19, 55, 49, 31], through post-processing the decisions [20], through adversarial learning [57, 7, 42]. These works generally assume the protected attribute information is known and thus the fairness metrics can be directly optimized. However, in many real world applications the protected attribute information might be missing or is very sparse.

**Fairness without demographics:**  Some works address this *approximately* by using proxy features [18] or assuming that the attribute is slightly perturbed [3]. However, using proxies can in itself be prone to estimation bias [29, 10]. Multiple works have explored addressing limited demographic data with transfer learning [11, 42, 47]. For example, Coston et al. [11] focus on domain adaptation of fairness in settings where the group labels are known for either source or target dataset. Mohri et al. [43] consider an agnostic federated learning, wherein given training data over $K$ clients with unknown sampling distributions, the model aims to learn mixture coefficient weights that optimize for a worst-case target distribution over these $K$ clients. Creager et al. [12] propose an invariant risk minimization approach for domain generalization where environment partitions are not provided.

An interesting line of work tackles this problem by relying on trusted third parties that collect and store protected-data necessary for incorporating fairness. They generally assume that the ML model has access to the protected-features, albeit in encrypted form via secure multi-party computation [50, 34, 25], or in a privacy preserving form by employing differentially private learning [50, 27].

As mentioned earlier, the work closest to ours is *DRO* [21], which uses techniques from distributionally robust optimization to achieve Rawlsian Max-Min fairness without access to demographics. However, a key difference between *DRO* and *ARL* is the type of groups identified by them: DRO considers any worst-case distribution exceeding a given size $\alpha$ as a potential protected group. Concretely, given a lower bound on size of the smallest protected group, say $\alpha$, *DRO* optimizes for improving the worst-case performance of any set of examples exceeding size $\alpha$. In contrast, our work relies on a notion of computational-identifiability.

**Computational-Identifiability:**  Related to our algorithm, a number of works [33, 35, 23, 36] address intersectional fairness by optimizing for group fairness between all computationally identifiable groups in the input space. While the perspective of learning over computationally identifiable groups is similar, they differ from us in that they assume the protected group features are available in their input space, and that they aim to minimize the *gap* in utility across groups via regularization.

**Modeling Technique Inspirations:**  In terms of technical machinery, our proposed ARL approach draws inspiration from a wide variety of prior modeling techniques. Re-weighting [28, 26, 41] is a popular paradigm typically used to address problems such as class imbalance by upweighting examples from minority class. Adversarial learning [17, 2, 44, 51] is typically used to train a model to be robust with respect to adversarial examples. Focal loss [39] encourages the learning algorithm to focus on more *difficult* examples by up-weighting examples proportionate to their losses. Domain adaptation work requires a model to be robust and generalizable across different domains, under either covariate shift [53, 48] or label shift [40].

## 3   Model

We now dive into the precise problem formulation and our proposed modeling approach.

### 3.1   Problem Formulation

In this paper we consider a binary classification setup (though the approach can be generalized to other settings). We are given a training dataset consisting of $n$ individuals $\mathcal{D} = \{(x_i, y_i)\}_{i=1}^{n}$ where

$x_i \sim X$ is an $m$ dimensional input vector of non-protected features, and $y_i \sim Y$ represents a binary class label. We assume there exist $K$ protected groups where for each example $x_i$ there exists an unobserved $s_i \sim S$ where $S$ is a random variable over $\{k\}_{k=0}^K$. The set of examples with membership in group $s$ given by $\mathcal{D}_s := \{(x_i, y_i) : s_i = s\}_{i=1}^n$. Again, we do not observe distinct set $\mathcal{D}_s$ but include the notation for formulation of the problem. To be more precise, we assume that protected groups $S$ are unobserved attributes not available at training or inference times. However, we will frame our definition and evaluation of fairness in terms of groups $S$.

**Problem Definition** Given dataset $\mathcal{D} \in X \times Y$, but *no observed protected group memberships $S$, e.g., race or gender*, learn a model $h_\theta : X \to Y$ that is fair to groups in $S$.

A natural next question is: what is a "fair" model? As in DRO [21], we follow the Rawlsian *Max Min* fairness principle of distributive justice [46]: we aim to maximize the minimum utility $U$ a model has across all groups $s \in S$ as given by Definition 1. Here, we assume that when a model predicts an example correctly, it increases utility for that example. As such $U$ can be considered any one of standard accuracy metrics in machine learning that models are designed to optimize for.

**Definition 1 (Rawlsian Max-Min Fairness)** *Suppose $H$ is a set of hypotheses, and $U_{\mathcal{D}_s}(h)$ is the expected utility of the hypothesis $h$ for the individuals in group $s$, then a hypothesis $h^*$ is said to satisfy Rawlsian* Max-Min *fairness principle [46] if it maximizes the utility of the worst-off group, i.e., the group with the lowest utility.*

$$h^* = \arg \max_{h \in H} \min_{s \in S} U_{\mathcal{D}_s}(h) \tag{1}$$

In our evaluation in Section 4, we use AUC as a utility metric, and report the minimum utility over protected groups $S$ as AUC(min).

## 3.2 Adversarial Reweighted Learning

Given this fairness definition and goal, how do we achieve it? As with traditional machine learning, most utility/accuracy metrics are not differentiable, and instead convex loss functions are used. The traditional ML task is to learn a model $h$ that minimizes the loss over the training data $\mathcal{D}$:

$$h^*_{\text{avg}} = \arg \min_{h \in H} L_{\mathcal{D}}(h) \tag{2}$$

where $L_{\mathcal{D}}(h) = \mathbb{E}_{(x_i, y_i) \sim \mathcal{D}}[\ell(h(x_i), y_i]$ for some loss function $\ell(\cdot)$ (e.g., cross entropy).

Therefore, we take the same perspective in turning Rawlsian Max-Min Fairness as given in Eq. (1) into a learning objective. Replacing the expected utility with an appropriate loss function $L_{\mathcal{D}_s}(h)$ over the set of individuals in group $s$, we can formulate our fairness objective as:

$$h^*_{\max} = \arg \min_{h \in H} \max_{s \in S} L_{\mathcal{D}_s}(h) \tag{3}$$

where $L_{\mathcal{D}_s}(h) = \mathbb{E}_{(x_i, y_i) \sim \mathcal{D}_s}[\ell(h(x_i), y_i]$ is the expected loss for the individuals in group $s$.

**Minimax Problem:** Similar to Agnostic Federal Learning (AFL) [43], we can formulate the Rawlsian *Max-Min Fairness* objective function in Eq. (3) as a zero-sum game between two players $\theta$ and $\lambda$. The optimization comprises of $T$ game rounds. In round $t$, player $\theta$ learns the best parameters $\theta$ that minimizes the expected loss. In round $t+1$, player $\lambda$ learns an assignment of weights $\lambda$ that maximizes the weighted loss.

$$J(\theta, \lambda) := \min_\theta \max_\lambda L(\theta, \lambda) = \min_\theta \max_\lambda \sum_{s \in S} \lambda_s L_{\mathcal{D}_s}(h)$$

$$= \min_\theta \max_\lambda \sum_{i=0}^n \lambda_{s_i} \ell(h(x_i), y_i) \tag{4}$$

To derive a concrete algorithm we need to specify how the players pick $\theta$ and $\lambda$. For the $\theta$ player, one can use any iterative learning algorithm for classification tasks. For player $\lambda$, if the group memberships were known, the optimization problem in Eq. 4 can be solved by projecting $\theta$ on a probability simplex over $S$ groups given by $\lambda = \{[0, 1]^S : \|\lambda\| = 1\}$ as in AFL [43]. Unfortunately, for us, because we do not observe $S$ we cannot directly optimize this objective as in AFL [43].

DRO [21] deals with this by effectively setting weights $\lambda_i$ based on $\ell(h(x_i), y_i)$ to focus on the largest errors. Instead, we will leverage the concept of *computationally-identifiable subgroups* [23]. Given a family of binary functions $\mathcal{F}$, we say that a subgroup $S$ is computationally-identifiable if there is a function $f : X \times Y \rightarrow \{0, 1\}$ in $\mathcal{F}$ such that $f(x, y) = 1$ if and only if $(x, y) \in S$.

Building on this definition, we define $f_\phi : X \times Y \rightarrow [0, 1]$ to be an adversarial neural network parameterized by $\phi$ whose task, implicitly, is to identify regions where the learner makes significant errors, e.g. regions $Z := \{(x, y) : \ell(h(x), y) \geq \epsilon\}$. Since our adversarial network $f_\phi$ is not a binary classifier, we aren't explicitly specifying $\epsilon$ but rather training so that $f_\phi$ returns a higher value in higher loss regions. The adversarial examples weights $\lambda_\phi : f_\phi \rightarrow \mathbb{R}$ can then be defined by appropriately rescaling $f_\phi$ to put a high weight on regions with a high likelihood of errors, encouraging $h_\theta$ to improve in these regions. Rather than explicitly enforce a binary set of weights, as would be implied by the original definition of computational identifiability, our adversary uses a sigmoid activation to map $f_\phi(x, y)$ to [0,1]. While this does not explicitly enforce a binary set of weights, we empirically observe that the rescaled weights $\lambda_\phi(x, y)$ results in the weights clustering in two distinct regions as we see in Fig. 5 (with low weights near 1 and high weights near 4) .

**ARL Objective:** We formalize this intuition, and propose an Adversarially Reweighted Learning approach, called *ARL*, which considers a minimax game between a *learner* and *adversary*: Both *learner* and *adversary* are learnt models, trained alternatively. The *learner* optimizes for the main classification task, and aims to learn the best parameters $\theta$ that minimizes expected loss. The *adversary* learns a function mapping $f_\phi : X \times Y \rightarrow [0, 1]$ to *computationally-identifiable* regions with high loss, and makes an adversarial assignment of weight vector $\lambda_\phi : f_\phi \rightarrow \mathbb{R}$ so as to maximize the expected loss. The *learner* then adjusts itself to minimize the adversarial loss.

$$J(\theta, \phi) = \min_\theta \max_\phi \sum_{i=1}^{n} \lambda_\phi(x_i, y_i) \cdot \ell_{ce}(h_\theta(x_i), y_i) \tag{5}$$

If the adversary was perfect it would *adversarially* assign all the weight ($\lambda$) on the computationally-identifiable regions where learner makes significant errors, and thus improve learner performance in such regions. It is worth highlighting that, the design and complexity of the adversary model $f_\phi$ plays an important role in controlling the granularity of *computationally-identifiable* regions of error. More expressive $f_\phi$ leads to finer-grained upweighting but runs the risk of overfitting to outliers. While any differentiable model can be used for $f_\phi$, we observed that for the small academic datasets used in our experiments, a *linear* adversary performed the best (further implementation details follow).

Observe that without any constraints on $\lambda$ the objective in Eq. 5 is ill-defined. There is no finite $\lambda$ that maximizes the loss, as an even higher loss could be achieved by scaling up $\lambda$. Thus, it is crucial that we constrain the values $\lambda$. In addition, it is necessary that $\lambda_i \geq 0$ for all $i$, since minimizing the negative loss can result in unstable behaviour. Further, we do not want $\lambda_i$ to fall to 0 for any examples, so that all examples can contribute to the training loss. Finally, to prevent exploding gradients, it is important that the weights are normalized across the dataset (or current batch). In principle, our optimization problem is general enough to accommodate a wide variety of constraints. In this work we perform a normalization step that rescales the adversary $f_\phi(x, y)$ to produce the weights $\lambda_\phi$. We center the output of $f_\phi$ and add 1 to ensure that all training examples contribute to the loss.

$$\lambda_\phi(x_i, y_i) = 1 + n \cdot \frac{f_\phi(x_i, y_i)}{\sum_{i=1}^{n} f_\phi(x_i, y_i)}$$

**Implementation:** In the experiments presented in Section 4, we use a standard feed-forward network to implement both *learner* and *adversary*. Our model for the learner is a fully connected two layer feed-forward network with 64 and 32 hidden units in the hidden layers, with *ReLU* activation function. While our adversary is general enough to be a deep network, we observed that for the small academic datasets used in our experiments, a *linear* adversary performed the best. Fig. 2 summarizes the computational graph of our proposed ARL approach.

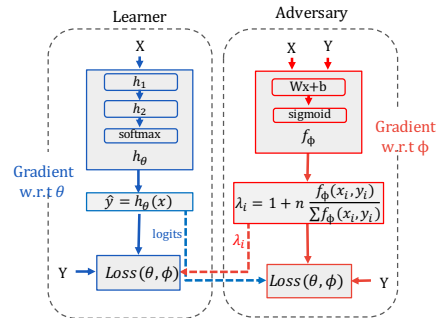

Figure 2: ARL Computational Graph

# 4 Experimental results

We now demonstrate the effectiveness of our proposed *ARL* approach through experiments over three real datasets[3] well used in the fairness literature: (i) Adult [45]: income prediction (ii) LSAC [52]: law school admission and (iii) COMPAS [1]: recidivism prediction.

**Evaluation Metrics:** We choose AUC (area under the ROC curve) as our utility metric as it is robust to class imbalance, i.e., unlike *Accuracy* it is not easy to receive high performance for trivial predictions. Further, it encompasses both *FPR* and *FNR*, and is threshold agnostic.

To evaluate fairness we stratify the test data by groups, compute AUC per protected group $s \in S$, and report (i) AUC(min): minimum AUC over all protected groups, (ii) AUC(macro-avg): macro-average over all protected group AUCs and (iii) AUC(minority): AUC reported for the smallest protected group in the dataset. For all metrics higher values are better. Values reported are averages over 10 runs. Note that the protected features are removed from the dataset, and are not used for training, validation or testing. The protected features are only used to compute subgroup *AUC* in order to evaluate fairness.

**Setup and Parameter Tuning:** We use the same experimental setup, architecture, and hyper-parameter tuning for all the approaches. As our proposed *ARL* model has additional model capacity in the form of example weights $\lambda$, we increase the model capacity of the baselines by adding more hidden units in the intermediate layers of their DNN in order to ensure a fair comparison. Best hyper-parameter values for all approaches are chosen via grid-search by performing 5-fold cross validation optimizing for best *overall* AUC. We do not use subgroup information for training or tuning. *DRO* has a separate *fairness* parameter $\eta$. For the sake of fair comparison, we report results for two variants of DRO: (i) DRO, with $\eta$ tuned as detailed in their paper and (ii) DRO (auc) with $\eta$ tuned to achieve best overall AUC performance. Refer to Supplementary for further details. All results reported are averages across 10 independent runs (with different model parameter initialization).

**Main Results: Fairness without Demographics:** Our main comparison is with *DRO* [21], a *group-agnostic* distributionally robust optimization approach that optimizes for the worst-case subgroup. Additionally, we report results for the vanilla *group-agnostic Baseline*, which performs standard ERM with uniform weights. Tbl. 1 reports results based on average performance across runs, with the best average performance highlighted in bold. Detailed results for all protected groups and with standard deviations across runs are reported in the Supplementary (Tbl. 6, 7, and 8). We make the following key observations:

*ARL improves worst-case performance:* ARL outperforms DRO, and achieves best results for AUC (minority) for all datasets. We observe a 6.5 percentage point (pp) improvement over the baseline for Adult, 0.8 pp for LSAC, and 1.1 pp for COMPAS. Similarly, ARL shows 2 pp and 1 pp improvement in AUC (min) over baseline for Adult and LSAC datasets respectively. For COMPAS dataset there is no notable difference in performance over baseline, yet substantially better than DRO, which suffers a lot.

These results are inline with our observations on computational-identifiability of protected groups (Tbl. 4) and robustness to label bias (Fig 3b) in Section (§5). As we will later see, unlike Adult and LSAC datasets, protected-groups in COMPAS dataset are not

Table 1: Main results: ARL vs DRO

| dataset | method | AUC avg | AUC macro-avg | AUC min | AUC minority |
|---|---|---|---|---|---|
| Adult | Baseline | 0.898 | 0.891 | 0.867 | 0.875 |
| Adult | DRO | 0.874 | 0.882 | 0.843 | 0.891 |
| Adult | DRO (auc) | 0.899 | 0.908 | 0.869 | 0.933 |
| Adult | ARL | **0.907** | **0.915** | **0.881** | **0.942** |
| LSAC | Baseline | 0.813 | 0.813 | 0.790 | 0.824 |
| LSAC | DRO | 0.662 | 0.656 | 0.638 | 0.677 |
| LSAC | DRO (auc) | 0.709 | 0.710 | 0.683 | 0.729 |
| LSAC | ARL | **0.823** | **0.820** | **0.798** | **0.832** |
| COMPAS | Baseline | **0.748** | **0.730** | **0.674** | 0.774 |
| COMPAS | DRO | 0.619 | 0.601 | 0.572 | 0.593 |
| COMPAS | DRO (auc) | 0.699 | 0.678 | 0.616 | 0.704 |
| COMPAS | ARL | 0.743 | 0.727 | 0.658 | **0.785** |

computationally-identifiable. Further, ground-truth recidivism class labels in COMPAS dataset are known to be noisy [14]. We suspect that noisy data, and biased ground-truth training labels play a role in the subpar performance of *ARL* and *DRO* for COMPAS dataset as both these approaches are susceptible to performance degradation in the presence of noisy labels as they cannot differentiate between mistakes on correct vs noisy labels as we later illustrate in Section 5. Due to label noise, the training errors are not easily computationally-identifiable. Hence, ARL shows no notable performance

gain or loss. In contrast, we believe DRO is picking on noisy outlier in the dataset as high loss example, hence the substantial drop in DRO's performance.

*ARL improves overall AUC:* Further, in contrast to the general expectation in fairness approaches, wherein utility-fairness trade-off is implicitly assumed, we observe that for Adult and LSAC datasets *ARL* in fact shows $\sim 1$ pp improvement in AUC (avg) and AUC (macro-avg). This is because *ARL*'s optimization objective of minimizing maximal loss is better aligned with improving overall *AUC*.

**ARL vs Inverse Probability Weighting:** Next, to better understand and illustrate the advantages of ARL over standard re-weighting approaches, we compare ARL with inverse probability weighting (*IPW*)[26], which is the most common re-weighting choice used to address representational disparity problems. Specifically, IPW performs a weighted ERM with example weights set as $1/p(s)$ where $p(s)$ is the probability of observing an individual from group $s$ in the empirical training distribution. In addition to vanilla IPW, we also report results for a IPW variant with inverse probabilities computed jointly over protected-features $S$ and class-label $Y$ reported as IPW(S+Y). Tbl. 2 summarizes the results. We make following observations and key takeaways:

Firstly, observe that in spite of not having access to demographic features, *ARL* has comparable if not better results than both variants of the *IPW* on all datasets. This results shows that even in the absence of group labels, *ARL* is able to appropriately assign adversarial weights to improve errors for protected-groups.

Further, not only does ARL improve subgroup fairness, in most settings it even outperforms IPW, which has perfect knowledge of group membership. This result further highlights the strength of *ARL*. We observed that this is because unlike IPW, ARL does not equally upweight all examples from protected groups, but does so only if the model needs much more capacity to be classified correctly. We present evidence of this observation in Section 5.

Table 2: ARL vs Inverse Probability Weight

| dataset | method | AUC avg | AUC macro-avg | AUC min | AUC minority |
|---------|--------|---------|---------------|---------|--------------|
| Adult | IPW(S) | 0.897 | 0.892 | 0.876 | 0.883 |
| Adult | IPW(S+Y) | 0.897 | 0.909 | 0.877 | 0.932 |
| Adult | ARL | **0.907** | **0.915** | **0.881** | **0.942** |
| LSAC | IPW(S) | 0.794 | 0.789 | 0.772 | 0.775 |
| LSAC | IPW(S+Y) | 0.799 | 0.798 | 0.784 | 0.785 |
| LSAC | ARL | **0.823** | **0.820** | **0.798** | **0.832** |
| COMPAS | IPW(S) | **0.744** | **0.727** | **0.679** | 0.759 |
| COMPAS | IPW(S+Y) | 0.727 | 0.724 | 0.678 | 0.764 |
| COMPAS | ARL | 0.743 | **0.727** | 0.658 | **0.785** |

**ARL vs Group-Fairness Approaches:** While our fairness formulation is not the same as traditional group-fairness approaches, in order to better understand relationship between improving subgroup performance vs minimizing gap, we compare *ARL* with a group-fairness approach that aims for equal opportunity (EqOpp) [20]. Amongst many EqOpp approaches [6, 20, 54], we choose *Min-Diff* [6] as a comparison as it is the closest to *ARL* in terms of implementation and optimization. To ensure fair comparison we instantiate *Min-Diff* with similar neural architecture and model capacity as *ARL*. Further, as we are interested in performance for multiple protected groups, we add one *Min-Diff* loss term for each protected feature (sex and race). Details of the implementation are described in Supplementary §8. Tbl. 3 summarizes these results. We make the following observations:

*Min-Diff improves gap but not worst-off group:* True to its goal, Min-Diff decreases the FPR gap between groups: FPR gap on sex is between $0.02$ and $0.05$, and FPR gap on race is between $0.01$ and $0.19$ for all datasets. However, lower-gap between groups doesn't always lead to improved AUC for worst-off groups (observe AUC min and AUC minority). ARL substantially outperforms Min-Diff for Adult and COMPAS datasets, and achieves comparable performance on LSAC dataset.

Table 3: ARL vs Group-Fairness

| dataset | method | AUC avg | AUC macro-avg | AUC min | AUC minority |
|---------|--------|---------|---------------|---------|--------------|
| Adult | MinDiff | 0.847 | 0.856 | 0.835 | 0.863 |
| Adult | ARL | **0.907** | **0.915** | **0.881** | **0.942** |
| LSAC | MinDiff | **0.826** | **0.825** | **0.805** | **0.840** |
| LSAC | ARL | 0.823 | 0.820 | 0.798 | 0.832 |
| COMPAS | MinDiff | 0.730 | 0.712 | 0.645 | 0.748 |
| COMPAS | ARL | **0.743** | **0.727** | **0.658** | **0.785** |

This result highlights the intrinsic mismatch between fairness goals of group-fairness approaches vs the desire to improve performance for protected groups. We believe making models more inclusive by improving the performance for groups, not just decreasing the gap, is an important complimentary direction for fairness research.

*Utility-Fairness Trade-off:* Further, observe that Min-Diff incurs a 5 pp drop in overall AUC for Adult dataset, and 2 pp drop for COMPAS dataset. In contrast, as noted earlier *ARL* in-fact shows an improvement in overall AUC for Adult and LSAC datasets. This result shows that unlike Min-Diff (or group fairness approaches in general) where there is an explicit utility-fairness trade-off, *ARL*

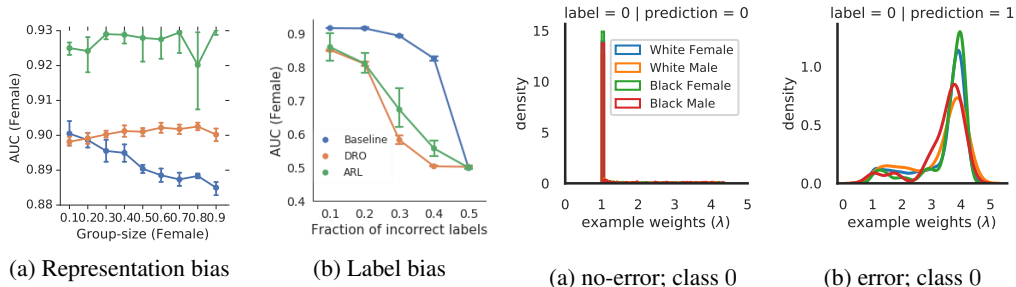

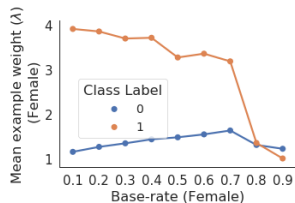

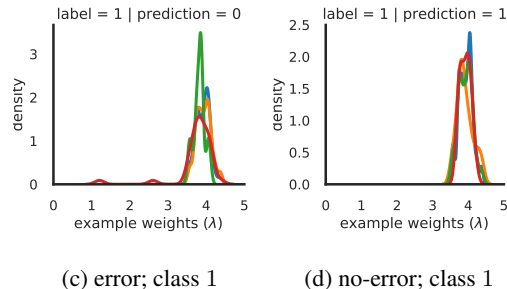

| (a) Representation bias | (b) Label bias |

Figure 3: Robustness to dataset biases.

(a) no-error; class 0     (b) error; class 0

Figure 4: Base-rate vs $\lambda$.

(c) error; class 1     (d) no-error; class 1

Figure 5: Example weights learnt by ARL.

achieves a better pareto allocation of overall and subgroup AUC performance. This is because the goal of *ARL*, which explicitly strives to improve the performance for protected groups is aligned with achieving better overall utility.

## 5 Analysis

Next, we conduct analysis to gain insights into *ARL*.

**Are groups computationally-identifiable?** We first test our hypothesis that unobserved protected groups $S$ are correlated with observed features $X$ and class label $Y$. Thus, even when they are unobserved, they can be computationally-identifiable. We test this hypothesis by training a predictive model to infer $S$ given $X$ and $Y$. Tbl. 4 reports the predictive accuracy of a linear model.

We observe that Adult and LSAC datasets have significant correlations with unobserved protected groups, which can be adversarially exploited to computationally-identify protected-groups. In contrast, for COMPAS dataset the protected-groups are not as computationally-identifiable [4]. As we saw earlier in Tbl. 1 (§4) these results align with ARL showing no gain or loss for COMPAS dataset, but improvements for Adult and LSAC.

Table 4: Identifying groups

|      | Adult | LSAC | COMPAS |
|------|-------|------|--------|
| Race | 0.90  | 0.94 | 0.61   |
| Sex  | 0.84  | 0.58 | 0.78   |

**Robustness to training distributions:** In this experiment, we investigate robustness of ARL and DRO approaches to training data biases [8], such as bias in group sizes (*representation-bias*) and bias due to noisy or incorrect ground-truth labels (*label-bias*). We use the Adult dataset and generate several semi-synthetic training sets with worst-case distributions (e.g., few training examples of "female" group) by sampling points from original training set. We then train our approaches on these worst-case training sets, and evaluate their performance on a fixed untainted original test set.

Concretely, to replicate *representation-bias*, we vary the fraction of female examples in training set by under/over-sampling female examples from training set. Similarly, to replicate *label-bias*, we vary fraction of incorrect labels by flipping ground-truth class labels uniformly at random for a fraction of training examples. In all experiments, training set size remains fixed. To mitigate the randomness in data sampling and optimization processes, we repeat the process 10 times and report results on a fixed untainted original test set (e.g., without adding label noise). Fig. 3 reports the results. In the interest of space, we limit ourselves to the protected-group "Female". For each training setting shown

on X-axis, we report the corresponding AUC for Female subgroup on Y-axis. The vertical bars in the plot are confidence intervals over 10 runs. We make the following observations:

*Representation Bias:* Both DRO and ARL are robust to the representation bias. ARL clearly outperforms DRO and *baseline* at all points. Surprisingly, we see a drop in AUC for *baseline* as the group-size increases. This is an artifact of having fixed training data size. As the fraction of female examples increases, we are forced to oversample female examples and downsample male examples; this leads to a decreases in the information present in training data and in turn leads to a worse performing model. In contrast, *ARL* and *DRO* cope better with this loss of information.

*Label Bias:* This experiment sheds interesting insights on the benefits of *ARL* over *DRO*. Recall that both approaches aim to focus on worst-case groups, however they differ in how these "groups" are formed. DRO is guaranteed to focus on worst-case risk for *any* group in the data exceeding size $\alpha$. In contrast, ARL would only improve the performance for groups that are computationally-identifiable over $(x, y)$.

We performed this experiment by setting DRO hyperparameter $\alpha$ to 0.2. We observed that while the fraction of incorrect ground truth class labels (i.e., outliers) is less than 0.2, the performance of ARL and DRO is nearly the same. As the fraction of outliers in the training set exceeds 0.2 we observe that DRO's performance drops substantially. These results highlight that, as expected both *ARL* and *DRO* are sensitive to label bias (as they aim to up-weight examples with prediction error but cannot distinguish between true and noisy labels). However, as noted by Hashimoto et al. [21], *DRO* exhibits a stronger trade-off between robustness to label bias and fairness.

**Are learnt example weights meaningful?** Next, we investigate if the example weights learnt by *ARL* are meaningful through the lense of training examples in the Adult dataset. Fig. 5 visualizes the example weights assigned by *ARL* stratified into four quadrants of a confusion matrix. Each subplot visualizes the learnt weights $\lambda$ on x-axis and their corresponding density on y-axis. We make following observations:

*Misclassified examples are upweighted:* As expected, misclassified examples are upweighted (in Fig. 5b and 5c), whereas correctly classified examples are not upweighted ( in Fig. 5a). Further, we observe that even though this was not our original goal, as an interesting side-effect *ARL* has also learnt to address class imbalance problem in the dataset. Recall that our Adult dataset has class imbalance, and only 23% of examples belong to class 1. Observe that, in spite of making no errors *ARL* assigns high weights to all class 1 examples as shown in Fig. 5d (unlike in Fig. 5a where all class 0 example have weight 1).

*ARL adjusts weights to base-rate.* We smoothly vary the base-rate of female group in training data (i.e., we synthetically control fraction of female examples with class label 1 in training data). Fig. 4 visualizes training data base-rate on x-axis and mean example weight learnt for the subgroup on y-axis. Observe that at female base-rate 0.1, i.e., when only 10% of female training examples belong to class 1, the mean weight assigned for examples in class 1 is significantly higher than class 0. As base-rate increases, i.e., as the number of class 1 examples increases, ARL correctly learns to decrease the weights for class 1 examples, and increases the weights for class 0 examples. These insights further explain the reason why *ARL* manages to improve overall AUC.

## 6  Conclusion

Improving model fairness without directly observing protected features is a difficult and under-studied challenge for putting machine learning fairness goals into practice. The limited prior work has focused on improving model performance for any worst-case distribution, but as we show this is particularly vulnerable to noisy outliers. Our key insight is that when improving model performance for worst-case groups, it is valuable to focus the objective on *computationally-identifiable* regions of errors i.e., regions of the input and label space with significant errors. In practice, we find *ARL* is better at improving AUC for worst-case protected groups across multiple dataset and over multiple types of training data biases. As a result, we believe this insight and the *ARL* method provides a foundation for how to pursue fairness without access to demographics.

**Acknowledgements:** The authors would like to thank Jay Yagnik for encouragement along this research direction and insightful connections that contributed to this work. We thank Alexander D'Amour, Krishna Gummadi, Yoni Halpern, and Gerhard Weikum for helpful feedback that improved the manuscript. This work was partly supported by the ERC Synergy Grant 610150 (imPACT).

# 7 Broader Impact

Any machine learning system that learns from data runs the risk of introducing unfairness in decision making. Recent research [19, 5, 30, 9] has identified fairness concerns in several ML systems, especially toward protected groups that are under-represented in the data. Thus, alongside the technical advancements in improving ML systems it crucial that we also focus on ensuring that they work for everyone.

One of the key practical challenges in addressing unfairness in ML systems is that most methods require access to protected demographic features, placing fairness and privacy in tension. In this work we work toward addressing these important challenges by proposing a new training method to improve worst-case performance of protected groups, in the absence of protected group information in the datasets. One limitation of methods in this space, including ours, is the difficulty of evaluating their effectiveness when we don't have demographics in a real application. Therefore, while we think developing better debiasing methods is crucial, there remains further challenges in evaluating them.

Further, this work relies on the assumption that protected groups are computationally-identifiable. However, if there were no signal about protected groups in the remaining features $X$ and class labels $Y$, we cannot make any statements about improving the model for protected groups. Similarly, we observe that when the ground truth labels in the training dataset are noisy, the performance of ARL drops. Looking forward, we believe further research is needed to validate how the proposed approach can remain effective in a wide variety of real world applications beyond the datasets we have studied in this work.

## Footnotes

*This work was conducted while the author was an intern at Google Research, Mountain View.

[2] Creditors may not request or collect information about an applicant's race, color, religion, national origin, or sex. Exceptions to this rule generally involve situations in which the information is necessary to test for compliance with fair lending rules. [CFBP Consumer Law and Regulations, 12 CFR §1002.5]

[3]Key characteristics of the datasets, including a list of all the protected groups are in Supplementary (Tbl. 9)

[4]While we did not perform a formal study on this, we observed that the adversarial predictive accuracy for race in COMPAS is 0.61 is consistent with prior work [37]. We believe that there are demographic signals in the data, but they are not strong enough to predict groups well.

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
