[Supplementary Material]

# 8    Supplementary Material

## 8.1    Additional Results: Significance of inputs to the Adversary

Our proposed adversarially reweighted learning *ARL* approach is flexible and generalizes to many related works by varying the inputs to the adversary. For instance, if the domain of our adversary $f_\phi(.)$ was $S$, i.e., it took only protected features $S$ as input, the sets $Z$ computationally-identifiable by the adversary boil down to an exhaustive cross over all the protected features $S$, i.e., $Z \subseteq 2^S$. Thus, our *ARL* objective in Eq. 5 would reduce to being very similar to the objective of fair agnostic federated learning by Mohri et al. [43]: to minimize the loss for the worst-off group amongst all *known* intersectional subgroups.

In this experiment, we further gain insights into our proposed adversarially re-weighting approach by comparing a number of variants *ARL*:

- ARL (adv: X+Y) : vanilla *ARL* where the adversary takes non-protected features $X$ and class label $Y$ as input.
- *ARL (adv: S)*: variant of *ARL* where the adversary takes only protected features $S$ as input.
- *ARL (adv: S+Y)*: variant of *ARL* with access to protected features $S$ and class label $Y$ as input.
- *ARL(adv: X+Y+S)*: variant of *ARL* where the adversary takes all features $X + S$ and class label $Y$ as input.

A summary of results is reported in Tbl. 5. We make the following observations:

*Group-agnostic ARL is competitive:* Firstly, observe that contrary to general expectation our vanilla ARL without access to protected groups $S$, i.e., *ARL (adv: X+Y)* is competitive, and its results are comparable with ARL variants with access to protected-groups $S$ (except in the case of COMPAS dataset as observed earlier). These results highlight the strength of ARL as an approach achieve fairness without access to demographics.

*Access to class label $Y$ is crucial:* Further, we observe that variants with class label ($Y$) generally outperform variants without class label. For instance, for *ARL(S+Y)* has higher AUC than *ARL(S)* for all groups across all datasets. Especially for Adult and LSAC datasets, which are known to have class imbalance problem (observe base-rate in Tbl.9). A similar trend was observed for *IPW(S)* vs *IPW(S+Y)* in Tbl.2 (§4). This is expected and can be explained as follows: variants without access to class label $Y$ such as *ARL(S)* are forced to give the same weight to both positive and negative examples of a group, As a consequence, they do not cope well with differences in base-rates, especially across groups, as they cannot treat majority and minority class differently.

*Blind Fairness:* Finally, in this work, we operated under the assumption that protected features are not available in the dataset. However, in practice there are scenarios where protected features $S$ are available in the dataset, however, we are *blind* to them. More concretely, we do not know a priori which subset of features amongst all features $X + S$ might be candidates for protected groups $S$. Examples of this setting include scenarios wherein a number of demographics features (e.g., age, race, sex) are present in the dataset. However, we do not known which subgroup(s) amongst all intersectional groups (given by the cross-product over demographic features) might need potential fairness treatment.

Table 5: A comparison of variants of ARL

| dataset | method | AUC | AUC macro-avg | AUC min | AUC minority |
|---|---|---|---|---|---|
| Adult | Baseline | 0.898 | 0.891 | 0.867 | 0.875 |
| Adult | ARL (adv: S) | 0.900 | 0.894 | 0.875 | 0.879 |
| Adult | ARL (adv: S+Y) | **0.907** | 0.907 | **0.882** | 0.907 |
| Adult | ARL (adv: X+Y+S) | **0.907** | 0.911 | 0.881 | **0.932** |
| Adult | ARL (adv: X+Y) | **0.907** | **0.915** | 0.881 | 0.942 |
| LSAC | Baseline | 0.813 | 0.813 | 0.790 | 0.824 |
| LSAC | ARL (adv: S) | 0.820 | 0.823 | 0.799 | **0.846** |
| LSAC | ARL (adv: S+Y) | 0.824 | **0.826** | 0.801 | 0.845 |
| LSAC | ARL (adv: X+Y+S) | **0.826** | 0.825 | **0.808** | 0.838 |
| LSAC | ARL (adv: X+Y) | 0.823 | 0.820 | 0.798 | 0.832 |
| COMPAS | Baseline | 0.748 | 0.730 | 0.674 | 0.774 |
| Compas | ARL (adv: S) | 0.747 | 0.729 | 0.675 | 0.768 |
| Compas | ARL (adv: S+Y) | 0.747 | **0.731** | **0.681** | 0.771 |
| Compas | ARL (adv: X+Y+S) | **0.748** | **0.731** | 0.673 | **0.778** |
| Compas | ARL (adv: X+Y) | 0.743 | 0.727 | 0.658 | 0.785 |

Our proposed ARL approach naturally generalizes to this setting as well. We observe that the performance of our ARL variant *ARL(adv: X+Y+S)* is comparable to the performance of *ARL(adv: Y+S)*. In certain cases (e.g., Adult dataset), access to remaining features $X$ even improves fairness. We believe this is because access to $X$ helps the adversary to make fine-grained distinctions amongst a subset of disadvantaged candidates in a given group $s \in S$ that need fairness treatment.

## 8.2 Omitted Tables

In Section 4 Our main comparison is with *DRO* [21], a *group-agnostic* distributionally robust optimization approach that optimizes for the worst-case subgroup. Additionally, we report results for the vanilla *group-agnostic Baseline*, which performs standard ERM with uniform weights. Tbl. 6 7 and 8 summarize the main results. We report AUC (mean $\pm$ std) for all protected groups in each dataset. Best values in each table are highlighted in bold.

Table 6: Adult: values in the table are AUC (mean $\pm$ std). Best results in bold.

| Method | AUC Overall | AUC White | AUC Black | AUC Male | AUC Female | AUC White Male | AUC White Female | AUC Black Male | AUC Black Female |
|---|---|---|---|---|---|---|---|---|---|
| Baseline | $0.898 \pm 0.0005$ | $0.894 \pm 0.0005$ | $0.919 \pm 0.0047$ | $0.882 \pm 0.0006$ | $0.882 \pm 0.0019$ | $0.879 \pm 0.0005$ | $0.881 \pm 0.0018$ | $0.914 \pm 0.004$ | $0.875 \pm 0.0241$ |
| DRO | $0.874 \pm 0.0007$ | $0.869 \pm 0.0007$ | $0.908 \pm 0.0022$ | $0.848 \pm 0.0009$ | $0.902 \pm 0.0012$ | $0.843 \pm 0.0009$ | $0.901 \pm 0.0014$ | $0.897 \pm 0.0034$ | $0.891 \pm 0.0015$ |
| DRO (auc) | $0.899 \pm 0.0003$ | $0.894 \pm 0.0004$ | $0.931 \pm 0.0014$ | $0.873 \pm 0.0006$ | $0.928 \pm 0.0008$ | $0.869 \pm 0.0007$ | $0.925 \pm 0.0008$ | $0.909 \pm 0.0022$ | $0.933 \pm 0.0008$ |
| ARL | $\mathbf{0.907 \pm 0.0009}$ | $\mathbf{0.903 \pm 0.001}$ | $\mathbf{0.932 \pm 0.0025}$ | $\mathbf{0.885 \pm 0.0011}$ | $\mathbf{0.93 \pm 0.0011}$ | $\mathbf{0.881 \pm 0.0012}$ | $\mathbf{0.927 \pm 0.0016}$ | $\mathbf{0.917 \pm 0.0024}$ | $\mathbf{0.942 \pm 0.0134}$ |

Table 7: LSAC: values in the table are AUC (mean $\pm$ std). Best results in bold.

| Method | AUC Overall | AUC White | AUC Black | AUC Male | AUC Female | AUC White Male | AUC White Female | AUC Black Male | AUC Black Female |
|---|---|---|---|---|---|---|---|---|---|
| Baseline | $0.813 \pm 0.0025$ | $0.799 \pm 0.0027$ | $0.828 \pm 0.0042$ | $0.816 \pm 0.0032$ | $0.808 \pm 0.0028$ | $0.805 \pm 0.0036$ | $0.79 \pm 0.0033$ | $0.824 \pm 0.0054$ | $\mathbf{0.83 \pm 0.0045}$ |
| DRO | $0.662 \pm 0.0002$ | $0.639 \pm 0.0003$ | $0.668 \pm 0.0005$ | $0.658 \pm 0.0003$ | $0.668 \pm 0.0003$ | $0.638 \pm 0.0004$ | $0.641 \pm 0.0003$ | $0.677 \pm 0.0013$ | $0.662 \pm 0.0$ |
| DRO (auc) | $0.709 \pm 0.0002$ | $0.687 \pm 0.0002$ | $0.733 \pm 0.0012$ | $0.709 \pm 0.0003$ | $0.71 \pm 0.0002$ | $0.691 \pm 0.0004$ | $0.683 \pm 0.0002$ | $0.729 \pm 0.0015$ | $0.737 \pm 0.0019$ |
| ARL | $\mathbf{0.823 \pm 0.004}$ | $\mathbf{0.811 \pm 0.0048}$ | $\mathbf{0.829 \pm 0.0153}$ | $\mathbf{0.829 \pm 0.0049}$ | $\mathbf{0.815 \pm 0.0066}$ | $\mathbf{0.819 \pm 0.0061}$ | $\mathbf{0.799 \pm 0.0068}$ | $\mathbf{0.832 \pm 0.0189}$ | $0.825 \pm 0.0173$ |

Table 8: COMPAS: values in the table are AUC (mean $\pm$ std). Best results in bold.

| Method | AUC Overall | AUC White | AUC Black | AUC Male | AUC Female | AUC White Male | AUC White Female | AUC Black Male | AUC Black Female |
|---|---|---|---|---|---|---|---|---|---|
| Baseline | $\mathbf{0.748 \pm 0.0035}$ | $\mathbf{0.713 \pm 0.0048}$ | $\mathbf{0.753 \pm 0.0049}$ | $\mathbf{0.749 \pm 0.0034}$ | $\mathbf{0.717 \pm 0.0086}$ | $0.724 \pm 0.0058$ | $\mathbf{0.674 \pm 0.01}$ | $\mathbf{0.738 \pm 0.005}$ | $0.774 \pm 0.011$ |
| DRO | $0.619 \pm 0.0049$ | $0.601 \pm 0.0058$ | $0.614 \pm 0.0045$ | $0.624 \pm 0.0043$ | $0.583 \pm 0.0098$ | $0.609 \pm 0.0055$ | $0.573 \pm 0.0154$ | $0.613 \pm 0.0048$ | $0.593 \pm 0.0094$ |
| DRO (auc) | $0.699 \pm 0.0049$ | $0.68 \pm 0.0055$ | $0.69 \pm 0.006$ | $0.704 \pm 0.0048$ | $0.655 \pm 0.0064$ | $0.696 \pm 0.0052$ | $0.616 \pm 0.013$ | $0.681 \pm 0.0068$ | $0.704 \pm 0.0119$ |
| ARL | $0.743 \pm 0.0027$ | $0.712 \pm 0.0051$ | $0.747 \pm 0.003$ | $0.745 \pm 0.0029$ | $0.714 \pm 0.0068$ | $\mathbf{0.725 \pm 0.0052}$ | $0.658 \pm 0.0093$ | $0.733 \pm 0.0037$ | $\mathbf{0.785 \pm 0.0074}$ |

## 8.3 Datasets and Pre-processing

**Datasets for Main Experiments:** We perform our experiments on three real-world, publicly available datasets, previously used in the literature on algorithmic fairness:

- Adult: The UCI Adult dataset [45] contains US census income survey records. We use the binarized "income" feature as the target variable for our classification task to predict if an individual's income is above $50k$.
- LSAC: The Law School dataset [52] from the law school admissions council's national longitudinal bar passage study to predict whether a candidate would pass the bar exam. It consists of law school admission records. We use the binary feature "isPassBar" as the raget variable for classification.
- COMPAS: The COMPAS dataset [1] for recidivism prediction consists of criminal records comprising offender's criminal history, demographic features (sex, race). We use the ground truth on whether the offender was re-arrested (binary) as the target variable for classification.

We transform all categorical attributes using one-hot encoding, and standardize all features vectors to have zero mean and unit variance. Python scripts for preprocessing the public datasets are open accessible along with the rest of the code of this paper.

Table 9: Description of datasets

| Dataset | Size | No. features | Protected features | Protected groups | Prediction task |
|---|---|---|---|---|---|
| Adult | 40701 | 15 | Race, Sex | {White, Black} $\times$ {Male, Female} | income $\geq$ 50k? |
| LSAC | 27479 | 12 | Race, Sex | {White, Black} $\times$ {Male, Female} | Pass bar exam? |
| COMPAS | 7215 | 11 | Race, Sex | {White, Black} $\times$ {Male, Female} | recidivate in 2 years? |

**Datasets for Synthetic Experiments:** In Section 5 we perform additional experiments on a number of semi-synthetic datasets to investigate robustness of ARL to training distributions. All the code to generate synthetic datasets is shared along with the rest of the code of this paper.

## 8.4 Baselines and Implementation

We compare our proposed approach *ARL* with the two naive baselines and one state-of-the-art approach. All the implementations are open accessible along with the rest of the code of this paper. All approaches have the same DNN architecture, optimizer and activation functions. As our proposed *ARL* model has additional model capacity in the form of example weights $\lambda$, in order to ensure fair comparison we increase the model capacity of the baselines by adding more hidden units in the intermediate layers of their DNN. Following are the implementation details:

- **Baseline**: This is a simple empirical risk minimization baseline with standard binary cross-entropy loss.
- **IPW**: This a naive re-weighted risk minimization approach with weighted binary cross-entropy loss. The weights are assigned to be inverse probability weights $1/p(s)$, where $p(s)$ is the . For a fair comparison, we train IPW with the same model as *ARL*, with fixed adversarial re-weighting. More concretely, rather than adversarially learning weights in a group-agnostic manner, the example weights ($\lambda$) are precomputed inverse probability weights $1/p(s)$. Additionally, we perform experiments on a variant of IPW called IPW (S+Y) with weights $1/p(s,y)$, where $1/p(s,y)$ is the joint probability of observing a data-point having membership to group $s$ and class label $y$ over empirical training distributions.
- **DRO**: This is a group-agnostic distributionally robust learning approach for fair classification. We use the code shared by [21], which is available at `https://worksheets.codalab.org/worksheets/0x17a501d37bbe49279b0c70ae10813f4c/`. We tune the hyper-parameters for *DRO* by performing grid search over the parameter space as reported in their paper.

## 8.5 Experimental Setup and Parameter Tuning

Each dataset is randomly split into 70% training and 30% test sets. On the training set, we perform a 5 fold cross validation to find the best hyper-parameters for each model (details follow). Once the hyperparameters are tuned, we use the second part as an independent test set to get an unbiased estimate of their performance. We use the same experimental setup, data split, and parameter tuning techniques for all the methods.

**Hyperparameter Tuning:** For each approach, we choose the best learning-rate, and batch size by performing a grid search over an exhaustive hyper parameter space given by batch size (32, 64, 128, 256, 512) and learning rate (0.001, 0.01, 0.1, 1, 2, 5). All the parameters are chosen via 5-fold cross validation by optimizing for best overall AUC.

In addition to batch size, and learning rate, DRO approach [21] has an additional *fairness* hyper-parameter $\eta$, which controls the performance for the worst-case subgroup. In their paper, the authors present a specific hyperparameter tuning approach to choose the best value for $\eta$. Hence for the sake of fair comparison, we report results for two variants of DRO: (i) DRO, original approach with $\eta$ tuned as detailed in their paper and (ii) DRO(auc) with $\eta$ tuned to achieve best overall AUC performance.

## 8.6 Reproducibility

All the datasets used in this paper are publicly available. The python and tensorflow implementation of proposed ARL approach, as well as scripts to generate synthetic datasets is available opensource at `https://github.com/google-research/google-research/tree/master/group_agnostic_fairness`.