[Reviews · NeurIPS 2020]

Review 1

Summary and Contributions: This paper proposes a fair classifier that minimizes the worse-case loss among different demographics (Rawlsian max-min fairness) without the knowledge of demographic information. The technique relies upon a reweighted example approach, while it is tailored for the interested fairness setting. Some experiments are provided that highlight the advantages and features of the proposed approach relative to equal-weighting baseline and DRO. Experimental analysis is offered to ease a deeper understanding of the approach.

Strengths: The paper addresses an important and practically-relevant problem. Overall it is well written, and in particular experimental analysis in Section 5 is insightful.

Weaknesses: 1. In the considered three datasets, DRO does not work well, even worse than the equal-weighting baseline. What about for other datasets (if any) in which the benefits of DRO are apparent? It may be the case in which there are very few noisy outliers. Even a synthetic dataset might be considered if available. If so, this needs to be extensively discussed with enough experimental results and potential analysis. 2. As shown in Table 1 (COMPAS), the proposed algorithm exhibits no or less gain, compared to the baseline, for scenarios in which group information is not reflected in (x,y) samples, as confirmed in Table 3. This reviewer wonders if there is a unified approach that performs best for all possible scenarios. 3. The employed reweighting approach is standard, while it offers great performance in the considered problem. 4. As demonstrated in Fig. 3(b), the proposed approach (together with DRO) is vulnerable to label bias. Wonder if it can be adopted to equip the robustness aspect against such poisoning.

Correctness: The claims supported by experiments and analysis look correct and the proposed method is reasonable.

Clarity: It is well written and easy to grasp the main ideas while it contains several typos.

Relation to Prior Work: See the 1st point left above in “Weaknesses”.

Reproducibility: Yes

Additional Feedback: ============== After rebuttal ============== I have read the authors' response as well as other reviewers' comments. My concerns are more or less addressed and I would stick to my initial rating.


Review 2

Summary and Contributions: The paper seeks to learn fair classifiers even when we do not have access to the sensitive group membership information at training. The main idea is to exploit the information in the known task labels and non-sensitive features as proxy information for the protected group information. Methodologically this is done by adversarially reweighing the samples to learn the task classifier. Experiments on small scale data demonstrate its effectiveness against one baseline. -------------Post Rebuttal Update---------- The authors addressed my concerns to an extent, but not fully. Nonetheless, the paper is interesting, so I will maintain my rating.

Strengths: + The paper addresses a very important and pertinent problem in fair machine learning. + The idea of relying on computational identifiability as opposed to distributional robustness. It does assume that the conditions necessary for computational identifiability are met. + Experimental results show that the method is effective and learn plausible weights. + The analysis of the results are helpful and aid in understanding the effects of the method.

Weaknesses: - A couple of existing work on this topic is not been discussed. See [1] and [2] below. - The computational identifiability criterion is largely driven by sample level weights predicted by the adversarial network. Unless the adversary can learn smooth functions there is no guarantee that the proposed approach will learn contiguous regions and not simply identify outliers. Is it possible that this is what is happening when the adversary is more complex than a linear layer? Is this why linear performed best? It might be helpful to visualize the identified regions, at least on toy datasets, with different capacity adversary networks. - Line 184 defines regions as l(h(x),y) > epsilon. But the loss function in Eq. 5 does not adhere to this. [1] Fair Transfer Learning with Missing Protected Attributes, AIES 2019 [2] Fairness Under Unawareness: Assessing Disparity When Protected Class Is Unobserved, FAT 2019.

Correctness: The main hypothesis and proposed adversarial reweighing method is sound. But there seems to be an inherent limitation/trade-off of adversarial reweighing approach, in terms of avoiding overfitting, that has not been sufficiently investigated.

Clarity: The paper is well written and clear for the most part. There are some missing details here and there. For example, \lambda_{s_i} in Eq .4 is never defined.

Relation to Prior Work: Some relevant work is missing and not discussed. See comment above.

Reproducibility: Yes

Additional Feedback:


Review 3

Summary and Contributions: This paper proposes a method for achieving a Rawlsian notion of fairness when the protected attributes are unobserved. This method targets reducing errors on computationally identifiable groups using adversarial reweighing.

Strengths: This paper tackles an important problem of interest to the NeurIPS community. The paper provides extensive empirical evaluation that give an in-depth look at the strengths and weaknesses. The experimental analysis of label bias and representation bias was very insightful. The paper is well-written.

Weaknesses: The paper could benefit from further theoretical work that would give guarantees on the method. -------- UPDATE after author response: I think the paper would benefit from a more in-depth analysis as to why the method does not work on COMPAS. Criminal justice is an application where we know that features like prior records and outcomes like arrest are racially biased. The fact that race in such a setting is not computationally identifiable seems not only surprising but also concerning since this suggests that in many applications of interest, the important protected attributes may not be identifiable according to this method.

Correctness: The claims and method appear correct.

Clarity: Yes, the paper is extremely well-written. The writing is very detailed, very clear, and easy to follow. The only place where I found the discussion was lacking was the observation that in COMPAS the protected groups are not computationally identifiable. This seems surprising and it would be helpful to give an explanation. In a given application, how should one understand whether computationally identifiable groups will include the groups they are concerned about?

Relation to Prior Work: Yes, the paper has a very solid related work section.

Reproducibility: Yes

Additional Feedback: It would be help to provide more explanation of the figures in the captions --e.g. explaining the shaded out colors in Figure 1 and what avg/macro avg/ minority mean in Table 1. It would be helpful early on to give some intuition about what are computationally identifiable groups; this only became clear to me after reading a few pages. It would also be helpful to give a definition environment when you formally define this on lines 180-181. Line 172 -- specifying that the weights are over S would be helpful. In Equation 5, what does the subscript "ce" mean? The claim that AUC (for ROC curves) is robust to class imbalance seems a bit strong. There's a perspective that when class imbalance is very strong, that PR curves are more informative than AUC (Davis, Jesse, and Mark Goadrich. "The relationship between Precision-Recall and ROC curves." Proceedings of the 23rd international conference on Machine learning. 2006.). Would it be possible to train the adversaries jointly (vs alternatively)? Why do we see the sharp dropoff in Figure 5? It would be useful to provide more detail in the broader impact section (that could reiterate some of the limitations raised earlier in the paper--for instance potential adverse consequences if there is label noise). When might this method miss important groups?

[Author Response · NeurIPS 2020]

Thank you for the thoughtful feedback and comments; we are delighted to see this positive response. All reviewers agree
that we tackle an important problem of interest to the NeurIPS community, and acknowledge our extensive/insightful
experimental analysis which shows improvement over state-of-the-art. The reviewers recognize that we present an
interesting idea of relying on computational-identifiability [R3] and our proposed adversarial reweighing method is
sound [R3] which is composed in a well-written paper [R1,R3,R4]. Thank you! We respond to your questions below:

[R1] *When are DRO's benefits apparent ...?* The main benefit of DRO is that it induces distributional robustness.
Consequently, DRO is guaranteed to focus on worst-case risk for *any* group in the data exceeding size $\alpha$. In contrast,
ARL would only improve the performance for groups that are computationally identifiable groups over (x,y). However,
in the fairness setting, this is not necessarily a benefit for DRO, as the DRO authors state in their ICML talk: "it leads to
a trade-off between the outlier resistance and fairness, and when the fraction of outliers in training data exceeds size $\alpha$
DRO is completely broken." Our experiments on label bias (Fig. 3(b)) shed some light on this. These experiments
were performed with DRO parameter $\alpha$ set to 0.2. We observe that while the fraction of incorrect ground truth class
labels (i.e., outliers) is less than 0.2, the performance of ARL and DRO is nearly the same. As the fraction of outliers in
the training set exceeds 0.2 we observe that DRO's performance drops significantly. We will revise related work, and
experiments section to reflect this more clearly.

[R1] *The employed reweighting approach is standard, while it offers great performance in the considered problem.*
While the reweighted optimization problem is standard, the key contribution of this paper is in the adversarial approach
of learning these weights over computable groups in (x,y). Our experiments on "ARL vs Inverse Probability Weighting"
(Tbl. 2) illustrate the advantages of ARL over standard re-weighting approaches.

[R1] *Is there a unified approach that performs best for all possible scenarios?* Thank you for your thoughtful comment.
This along with your other comment on equipping robustness ignited interesting discussions amongst the authors.
We think combining the strengths of computationally identifiable regions with distributional robustness would be an
interesting research direction to unify the strengths of the two approaches in future research.

[R3] *Regarding smoothness of computationally-identifiable regions, and avoiding overfitting to outliers:* Your obser-
vations, and understanding is correct. The design and complexity of the adversary model $f_\phi$ plays an important role
in controlling the granularity (and smoothness) of computationally-identifiable regions of error. More expressive $f_\phi$
lead to finer-grained upweighting but runs the risk of overfitting to outliers. We currently discuss this on line 200-202,
but for further clarity we will expand this discussion, and include experimental results on the relationship between the
adversary's complexity and the smoothness of the function learnt by the adversary. Further, recall that the adversary
example weight function $\lambda_\phi(x_i, y_i) : f_\phi(x_i, y_i) \to \mathbb{R}$ in Eq. 5 is over $f_\phi(x_i, y_i)$. Hence, the distribution of training
weights $\lambda_\phi(x_i, y_i)$ is a good indicator of the smoothness of the function $f_\phi$ learnt by the adversary, and a practical
heuristic to check if the model is overfitting to training outliers.

[R3] *Additional related work:* Thank you for pointing to relevant related work Coston et al. [AIES 2019] and Chen et
al. [FAccT 2019]. We will update our related work section. Coston et al. focus on domain adaptation of fairness in
settings where the group labels are known for either source or target dataset. Chen et al. investigate the reasons for bias
in estimating unfairness in the settings where proxies for group labels are used to evaluate model unfairness.

[R3] *Line 184 defines regions as $l(h(x), y) > \epsilon$. But the loss function in Eq. 5 does not adhere to this.* We intended to
use this notation from a descriptive perspective as a framing device to explain what we mean by a "significant error".
However, this is not necessary for optimization. We will update the model section to clarify this.

[R4] *COMPAS protected groups are not computationally-identifiable ... surprising:* While we did not perform a formal
study on this, we observed that the adversarial predictive accuracy for race in COMPAS is 0.61 is consistent with prior
work [1]. We believe that there are demographic signals in the data, but they are not strong enough to predict groups well.

[R4] *How should one understand if computationally-identifiable groups will include the protected groups?* Even
evaluating fairness without any group labels is very difficult[2], and as such with *zero* information this would be hard
to do. However, we expect in many scenarios practitioners have a variety of domain knowledge they can leverage.
For example, if groups labels were not available in training but for a small curated validation set or in an auxiliary
dataset, one could evaluate if the computationally identified groups include protected groups, e.g., by analyzing the
learnt weights (as in Fig. 4). Practioners might also have knowledge about their domain, e.g., skin tone is present in
images and heavily correlated with race. We will add a discussion on limitations of computational identifiability to the
paper, and update the broader impact sections to reflect this.

[R4] Thank you for helpful additional feedback. We will address all the comments by appropriately revising the text,
and reiterating some of the earlier limitations on label noise in broader impact sections for further clarity.

## Footnotes

[1] See Fig. 4 in "iFair: Learning individually fair data representations for algorithmic decision making. In ICDE 2019."

[2] See "Assessing algorithmic fairness with unobserved protected class using data combination. In FAccT 2020."


[Meta-Review · NeurIPS 2020]

Overall the reviewers agreed that the strengths of the paper outweighed its weaknesses and were in favor of acceptance. In preparing the paper for publication the authors should seek to address R3's outstanding concern regarding the results on the COMPAS data. Given that this data has come to be used as a benchmark, not being able to provide a clear explanation of why the approach does not work on COMPAS (why race is not computationally identifiable) is of critical importance.